# Effect of Co-Crosslinking Reaction on the Morphology of Octavinyl Polyhedral Oligomeric Silsesquioxane/Natural Rubber Composites

**DOI:** 10.3390/ijms26052001

**Published:** 2025-02-25

**Authors:** Guliang Fu, Mengyan Li, Xue Luo, Ziqing Tang, Rentong Yu, Jianhe Liao

**Affiliations:** School of Materials Science and Engineering, Hainan University, Haikou 570228, China; 13907504371@163.com (G.F.); limengyan133xx1539@163.com (M.L.); 19834407278@163.com (X.L.); 13700450240@163.com (Z.T.)

**Keywords:** natural rubber, polyhedral oligomeric silsesquioxanes, crystallization, co-crosslinking reaction, competitive reaction

## Abstract

Octavinyl polyhedral oligomeric silsesquioxane (OV-POSS) was synthesized and characterized by means of proton nuclear magnetic resonance (^1^H NMR) together with Fourier transform infrared spectroscopy (FT-IR). The nanocages were then introduced to natural rubber (NR) to afford organic–inorganic hybrid composites. Upon curing with dicumyl peroxide, the size of the dispersed phase was observed to decrease monotonically with increasing OV-POSS content, as depicted in scanning electron microscopy (SEM) images. This behavior differs significantly from that of analogous OV-POSS/NR vulcanizates cured with sulfur reported previously. To address this, the proportion of crosslinked OV-POSS was calculated using the results of FT-IR. The reaction enthalpy of the composites with different OV-POSS loading was recorded and analyzed by differential scanning calorimetry (DSC). A co-crosslinking reaction was suggested to play a crucial role in determining the morphologies of the composites. In addition, the agglomeration and crystallization of OV-POSS can also affect the morphologies of the composites.

## 1. Introduction

Since Williams identified that the chemical structure of caoutchouc coincides with that of isoprene, and Goodyear serendipitously discovered the crosslinking process of rubber [1,2], natural rubber (NR) has been accepted as one of the “enlighteners” in the history of polymer science and engineering. Unlike synthetic elastomers, NR is a sustainable biomass derived from the para rubber tree (Hevea brasiliensis), which contributes to achieving carbon neutrality [3]. Additionally, comprehensive performances such as low heat build-up, high tensile strength, and excellent elastic resilience of NR are distinguished from synthesized rubbers, which renders it a promising material for practical applications [4,5]. Natural rubber is still an integral part of elastomeric materials in today’s world and is used in a variety of applications [6]. However, the mechanical and thermal properties of NR still need to be improved. To achieve high-performance materials, multifarious inorganic fillers have been incorporated into NR using the convenient and powerful tool of composite design [7,8,9,10,11,12,13]. Zhuang et al. prepared a core–shell structure F-GA filler covalently bonded to NR and fabricated a three-dimensional layered structure F-GA/NR composite by vacuum-assisted filtration, which exhibited excellent mechanical properties and good electrical insulation [14]. Lakshmipriya and his colleagues combined octaphenyl (OP), POSS @ TESPT (TT), and multi-walled carbon nanotubes (MWCNT) @ POSS (VC) polyhedral oligomeric silsesquioxane (POSS) with natural rubber (NR) to prepare polymer matrix nanocomposites. The effects of the structure and concentration of nanofillers, together with the compatibility with the NR matrix, on various diffusion parameters were analyzed. Through the study of mechanical properties, the effect of fillers on chain flexibility was explored [15].

Cage-shaped silsesquioxane (POSS), among the smallest spherical silica nanoparticles with a precise structure (RSiO_3_/_2_)_n_ (where *n* = 8, 10, 12), has recently attracted wide interest due to its effectiveness in enhancing the performance of polymeric matrix composites, such as toughness, thermal stability, biocompatibility, flame retardancy, and so on [16]. Gorkem et al. proposed a new strategy for organic–inorganic hybrid networks by in situ type II photoinitiated polymerization of methyl methacrylate and diethanol amino-functionalized low-hybrid disiloxane (POSS-DEA) [17]. Grala and his colleagues used polypropylene (iPP), maleic anhydride functionalized PP (PP-g-MA), and amine functionalized POSS (aminopropyl heptaethyl isobutyl-POSS, ambPOSS; aminopropyl heptaethyl isooctyl-POSS, amoPOSS; or aminoethyl aminopropyl heptaethyl isobutyl-POSS, am2bPOSS) to graft POSS onto PP chains during reactive melt blending to obtain hybrids. The effects of POSS chemical structure and grafting degree on morphological characteristics and mechanical properties were studied [18]. In the area of rubber matrix composites reinforced by micro/nano inorganic particles, carbon black (silica) has been widely accepted as an effective candidate, especially for natural rubber. In the family of silica, POSS has been considered the smallest one and could play a positive role in view of its small size effect. Furthermore, OV-POSS, the most inexpensive POSS, can react with the double bonds of NR and weaken the immiscibility between this kind of silica and NR. In this sense, by tailoring functional groups (e.g., alkyl, alkylene, carboxyl, acrylate) tethered to the nanocages, composite properties can be finely tuned. Accordingly, targeted properties of elastomers can be achieved with the incorporation of non-reactive or reactive POSS [19,20]. At the same time, the relationship between composite properties and microstructure should be unveiled in depth.

In this contribution, octavinyl-polyhedral oligomeric silsesquioxane (OV-POSS) was synthesized and then introduced into natural rubber (NR) to prepare NR-based compounds (uncured samples). After curing, vulcanizates with different contents of POSS were obtained. The morphologies of the composites were observed by means of scanning electron microscopy (SEM). The proportion of OV-POSS crosslinked with the NR matrix was calculated using the results of Fourier transform infrared spectroscopy. Furthermore, the effect of reaction heat on the morphologies of the composites was discussed.

## 2. Results and Discussion

### 2.1. Synthesis of OV-POSS

The chemical structure of the synthesized OV-POSS was characterized by means of nuclear magnetic resonance spectroscopy (^1^H NMR), as depicted in Figure 1. The multiple peaks in the chemical shift range δ = 5.89–6.34 correspond to the vinyl hydrogen atoms of OV-POSS, while the singlet at δ = 7.26 is attributed to the chemical shift of CDCl_3_. No signal corresponding to the Si-OH group was observed in the ^1^H NMR spectrum, indicating that the hydrolysis and condensation reactions were complete. Therefore, it can be concluded that the OV-POSS was successfully synthesized. ^1^H NMR (400 MHz, Chloroform-d): δ 6.34–6.06 (m, 2H), 5.93 (dd, 1H).

The crystalline octavinyl-POSS was further confirmed by Fourier transform infrared (FTIR) spectroscopy. As shown in Figure 2, the Si-OH absorption band in the range of 950–810 cm^−1^ [21] is barely detectable, indicating the completion of the condensation reactions. The intense stretching vibration absorption band at 1109 cm^−1^ is characteristic of the Si-O-Si skeleton, while the absorption band at 582 cm^−1^ corresponds to the symmetric stretching vibration of the Si-O-Si bond. The absorption bands at 1604, 1406, and 1276 cm^−1^ are attributed to the stretching vibrations of the vinyl -CH_2_=CH_2_- bonds [22], and the signal at 3068 cm^−1^ is assigned to the stretching vibration of the vinyl C-H bond. Additionally, the signal at 970 cm^−1^ is ascribed to the out-of-plane bending vibration of CH groups in the vinyl -C=C- bonds. From these characteristic absorption bands, the structure of the target product can be well verified.

### 2.2. FT-IR Analysis of OV-POSS/NR Composites

Figure 3 displays the infrared spectra of OV-POSS/NR composites with varying OV-POSS content. To investigate the effect of crosslinking on morphologies, composites of uncured samples (compounds C0, C5, C10, C15, and C20) together with cured samples (vulcanizates V0, V5, V10, V15, and V20) were compared. The typical telescopic vibrational absorption of Si-O-Si bonds of OV-POSS appears at about 1109 cm^−1^ for the vulcanized NR composites with the addition of 5, 10, 15, and 20 portions of OV-POSS, and the typical characteristic bands of the Si-O-Si bonds shift to 1126 cm^−1^ for the uncured OV-POSS/NR compounds. It can be inferred that the packing states of the OV-POSS domains should be different before and after curing.

Following the crosslinking reaction, a decrease is observed in the intensity of the antisymmetric stretching vibration of -CH_2_=CH_2_-, while the out-of-plane bending vibration of CH groups of vinyl -CH_2_=CH_2_- bonds remains at 970 cm^−1^. In addition, there is a residual vinyl characteristic band at 1410 cm^−1^, indicating that not all the reactive groups of OV-POSS were involved in the co-crosslinking reaction, which may be due to the spatial site resistance of OV-POSS [23]. In addition, absorption bands at 1724 cm^−1^ and 1541 cm^−1^, which are attributed to the carbonyl groups of phospholipids and amide groups of proteins, respectively, can be detected in the NR-based composites [24]. A broad FTIR band at 3200–3450 cm^−1^, ascribed to amino groups of proteins, can also be detected.

To evaluate the state of OV-POSS domains in the vulcanizates after the co-crosslinking reaction quantitatively, the content of residual OV-POSS domains in the vulcanizates can be calculated according to the intensity ratio of the characteristic absorption bands at 1410 cm^−1^ of OV-POSS to the characteristic methyl group absorption bands at 1376 cm^−1^ of NR, given that all of the methyl groups of NR would remain unchanged before and after the crosslinking reaction and all the FTIR spectra were normalized. For this reason, the relationship between the content of residual OV-POSS domains and the intensity ratio of A_1410_/A_1376_ was explored using the OV-POSS/NR compounds. For the uncured compounds, a linear fitting was achieved by plotting the intensity ratio of A_1410_/A_1376_ against the content of residual OV-POSS in the NR-based compounds (C0–C20). For the purpose of accuracy, a differentiating analysis of the overlapping bands was conducted. The calibration curve shown in Figure 4 provides a method to determine the mole content of residual OV-POSS domains in an NR vulcanizate. The mole percent of residual OV-POSS domains in NR-based vulcanizates can be calculated as follows:Y_POSS_ = 1250.95 × (A_1410_/A_1376_) − 6.51(1)

In the crosslinking reaction initiated by the free radical initiator DCP, one portion of the vinyl groups of OV-POSS can react with the double bonds of NR, while the other portion of OV-POSS would be isolated owing to the spatial hindrance effect. By calculating the intensity ratio of A_1410_ and A_1376_, the mole content of residual OV-POSS domains in the vulcanizates was obtained, and the results are listed in Table 1. It can be found that the mole proportion of residual OV-POSS domains in the vulcanizates generally increases with the increase in the feed ratio in the first stage, which indicates that OV-POSS can undergo a co-crosslinking reaction with NR. It should be pointed out that the vulcanizate with the incorporation of 10 phr OV-POSS exhibits a co-crosslinking yield of 93.8%, which could result from the lowest spatial hindrance. However, the proportion of crosslinked OV-POSS decreases to only 28.7% for V15.

### 2.3. Morphologies of OV-POSS/NR Vulcanizates

The state of OV-POSS cages (isolated or crosslinked with NR) in the NR-based vulcanizates can consequently have an effect on the morphologies of the vulcanizates. In principle, the formation of an organic–inorganic hybrid structure resulting from the co-crosslinking reaction could facilitate the improvement of miscibility between OV-POSS and NR. Accordingly, OV-POSS could be dispersed uniformly in the NR matrix. Nevertheless, the self-packing of OV-POSS due to its high specific surface energy cannot be suppressed, and aggregation of the nanoscale domains will take place when the content of OV-POSS reaches a certain value. As can be observed in Figure 5, there are many round crystalline particles randomly distributed in the NR matrix. With the increase in OV-POSS content, the numerical density of the dispersed phase increases accordingly. However, the size of the dispersed phase shows a tendency to decrease, which is considerably different from the morphologies of sulfur-cured OV-POSS/NR vulcanizates we reported previously [25]. To further investigate this, microstructural details of the dispersed phase are further examined. As shown in Figure 6, a downward trend in the size of the dispersed phase can be obviously observed with an increase in the content of OV-POSS. For the vulcanizate containing 5 phr OV-POSS, aggregates with an aspect ratio of 90.01 μm to 60.97 μm can be detected. At the same time, the self-packing of OV-POSS can be clearly observed. As the content of OV-POSS increases, aggregates with diameters of 61 μm and 36 μm can be observed for V10 and V15, respectively. Furthermore, a rough interface between the dispersed phase and the NR matrix can be observed, which could be attributed to the co-crosslinking reaction between OV-POSS and NR. It should be pointed out that cubic crystals of OV-POSS with an edge length of 14 μm are embedded in the NR matrix for the vulcanizate incorporating 20 phr OV-POSS. In the same field of view, we found that the proportion of the dispersed phase area (considering particle number and size in the same scope) of V15 tends to be larger than that of others. Hence, the morphologies of the composites are considerably different along with the content of OV-POSS. In this system, the morphologies of the OV-POSS/NR vulcanizates were affected by the competitive “reaction” of crosslinking, crystallization, and particle agglomeration enthalpy change. For V5 and V10, the distance between neighboring particles is relatively larger. As a result, the OV-POSS could co-crosslink with NR more effectively during the OV-POSS diffusion process. When the loading of OV-POSS is high enough, as in V20, agglomeration and self-packing of OV-POSS could take place preferentially before the crosslinking network is created, thus forming some small-sized crystal particles. Given that the reactive groups within OV-POSS will take part in the crosslinking reaction of NR, the effect of reaction heat on morphologies should not be ignored. In addition, the exothermic reaction during agglomeration and crystallization of OV-POSS particles should also have an effect on the morphologies of the OV-POSS/NR vulcanizates. The reaction heat during the isothermal curing of the OV-POSS/NR composite materials should be tracked and analyzed.

### 2.4. Enthalpy of Reaction for OV-POSS/NR Compounds

This distinct phase-separation behavior can be interpreted in terms of the heat of the reaction. As shown in Figure 7, the samples were heated to 160 °C with a ramp rate of 50 K/min from room temperature and then kept at this temperature for 300 min. The reaction was completed after 200 min, which can be validated by the disappearance of the DSC exothermic peak after that time. The exothermic peak area of blank NR is generally smaller than those of the composites containing OV-POSS, indicating that OV-POSS can indeed participate in the crosslinking reaction. Moreover, the enthalpy of reaction is found to generally increase with the increase in OV-POSS content, as shown in Table 2. However, the exothermic enthalpy decreases to 2208 J/g for the compound comprising 15 phr OV-POSS. According to Table 1 and Figure 6, in this sample, only 28.7 mol% of OV-POSS was crosslinked to the NR matrix. Considering the loading of OV-POSS, about 4.305 phr of OV-POSS reacted with the vinyl groups of NR or OV-POSS itself. This judgment is also applicable for C20, even though the crystallization of OV-POSS would bring out phase change heat and result in a relatively higher enthalpy of 2368 J/g. From Table 2, it can be found that the reaction enthalpy during the isothermal crosslinking reaction is not linearly consistent with the content of crosslinked OV-POSS. From this point of view, heat release resulting from crystallization and particle agglomeration did play an important role in the morphologies of the OV-POSS/NR vulcanizates. In fact, it is fairly hard to draw a simple conclusion directly due to the complexity of competitive reactions. Kanbargi et al. [26] revealed the significant influence of the linker’s conformational degrees of freedom on the segmental dynamics and therefore the material’s properties. In their findings, the suppression of chain dynamics at the particle interface would be governed by chain stretching and flexibility once covalent bonds were formed between the polymer and rigid particles. In this sense, during the isothermal crosslinking reaction, a lower co-crosslinking degree will give rise to a slowing down of segmental dynamics. Accordingly, a percolated network of glassy or immobilized polymer bridges will be formed in the interphase, which will in turn reduce the tendency of particle agglomeration.

## 3. Materials and Methods

### 3.1. Materials

Natural rubber (NR) of grade V (cis-polyisoprene > 96 wt%) was supplied by Hainan Rubber Industry Group Corporation Jinzhu Factory Co. (Haikou, China). All other chemicals, including vinyltriethoxysilane (97%), acetone (AR), concentrated hydrochloric acid (12.0 M), dichloromethane (AR), tetrahydrofuran (THF), dicumyl peroxide (DCP, AR), kerosene (CP), and anhydrous ethanol (99.5%), were purchased from Sinopharm Chemical Reagent Co., Ltd. (Shanghai, China).

### 3.2. Methods

#### 3.2.1. Synthesis of OV-POSS

In a three-necked round-bottom flask, vinyltriethoxysilane (22 mL, 3.78 mmol), acetone (350 mL, 16.06 mmol), concentrated hydrochloric acid (12.38 mL), and deionized water (8 mL) were added sequentially. With the protection of a nitrogen atmosphere, the materials mentioned above were vigorously stirred and reacted at 45 °C for 40 h. Afterward, the solution was rotary evaporated in vacuo. The resulting white precipitates were filtered and rinsed with anhydrous ethanol, followed by drying in a vacuum oven at 60 °C for 6 h. The crude product was then recrystallized with dichloromethane, and 5.30 g of OV-POSS with a yield of 37.9% was obtained. The reaction scheme for the synthesis of OV-POSS is presented in Figure 8.

#### 3.2.2. Fabrication

As shown in Figure 9, 10.00 g of NR was first dissolved in 120 mL of kerosene. A 20 mL THF solution containing 0.5 g of DCP was then prepared to dissolve 0.5, 1.0, 1.5, and 2.0 g of OV-POSS, respectively. Afterward, each OV-POSS/DCP/THF solution was added dropwise to the NR/kerosene solution to achieve homogeneous compounds. Finally, the solutions were allowed to evaporate at room temperature for 5 days, and the compounds were further dried in a vacuum oven at 35 °C for 24 h.

The OV-POSS/NR vulcanizates were prepared by curing at 160 °C for 1 h. The curing conditions were determined based on the results of the rotorless vulcanization test. To investigate the proportion of OV-POSS crosslinked with the NR matrix, cured and uncured NR composites incorporating different contents of OV-POSS were prepared first. For brevity, the vulcanizate containing 15 phr of OV-POSS is abbreviated as V15 (vulcanizate15), and the uncured composite containing 15 phr of OV-POSS is named C15 (compound15). In this way, other samples are also abbreviated based on the OV-POSS content and compositing state.

### 3.3. Characterization

The ^1^H NMR spectra of the synthesized products were recorded using an AVANCE NEO 400 NMR spectrometer with deuterated chloroform (CDCl_3_) as the solvent at room temperature.

FTIR spectroscopy characterization was performed on OV-POSS (KBr pellet method) and OV-POSS/NR composites (attenuated total reflection mode, ATR) at room temperature using a Nicolet iS10 spectrometer (Bruker, Karlsruhe, Germany). The wavelength ranges were set as 400–4000 cm^−1^ for the KBr pellet method and 650–4000 cm^−1^ for ATR, respectively. A resolution of 4 cm^−1^ was applied to record the spectra. All spectra were normalized for further analysis.

The morphologies of the samples were observed using a SU8010 scanning electron microscope (Hitachi Ltd., Tokyo, Japan) at an accelerating voltage of 30 kV. All the samples were freeze-fractured in liquid nitrogen and subsequently gold-coated prior to the experiments.

The reaction heat of the OV-POSS/NR composites undergoing crosslinking reaction initiated by DCP was recorded using a DISCOVER DSC250 differential scanning calorimetry (DSC, TA, New Castle, DE, USA). Typically, a 10 mg OV-POSS/NR compound was heated to 160 °C at a ramping speed of 50 K/min. Subsequently, the crosslinking reaction of the compound was carried out at 160 °C. To ensure the completion of the reaction, isothermal heating was maintained for 1 h, and the enthalpy change was recorded by DSC.

## 4. Conclusions

In this work, OV-POSS was successfully synthesized and then introduced to NR. Compounds with different contents of OV-POSS were cured at 160 °C for 1 h. Morphologies of the vulcanizates were verified to be affected not only by OV-POSS content but also by the co-crosslinking reaction between OV-POSS and NR or OV-POSS itself. It was found that the size of the dispersed phase tended to decrease with the increase in OV-POSS content in the vulcanizates. At the same time, OV-POSS could be dispersed uniformly in the NR matrix. The proportion of crosslinked OV-POSS was calculated using the results of FTIR, and it was found that the molar ratio of residual OV-POSS in the vulcanizate generally increased with the increase in the feed ratio in the first stage, which indicated that OV-POSS could co-crosslink with NR. The co-crosslinking yield of the vulcanizate with 10 phr OV-POSS was 93.8%. Co-crosslinking reaction enthalpies recorded by DSC were considered to play a role in the changes in morphologies of the composites.

## Figures and Tables

**Figure 1 ijms-26-02001-f001:**
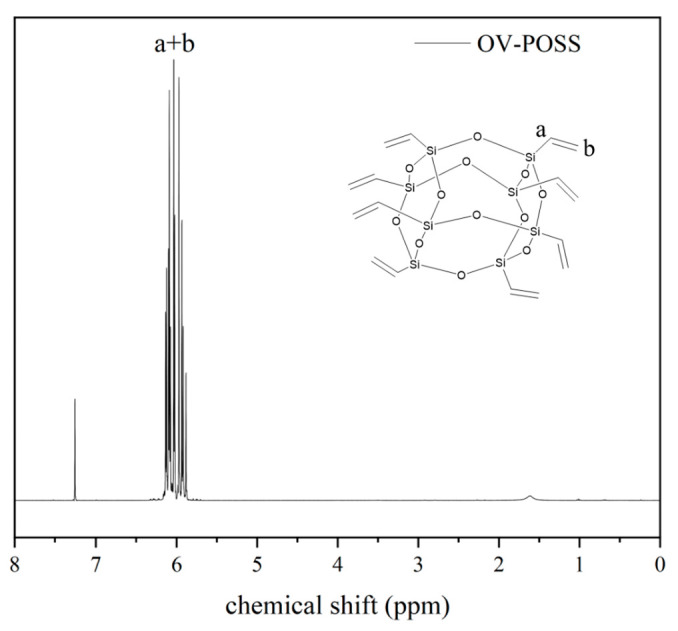
^1^H NMR spectrum of OV-POSS.

**Figure 2 ijms-26-02001-f002:**
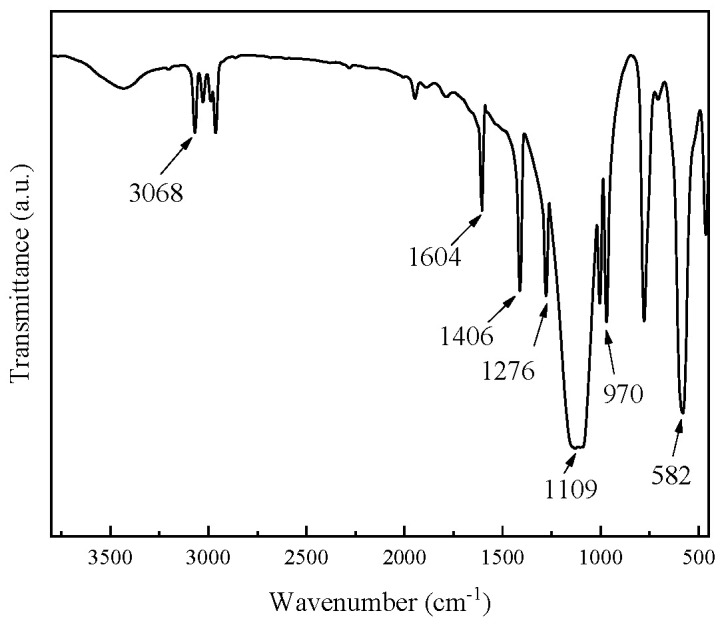
FTIR spectrum of OV-POSS.

**Figure 3 ijms-26-02001-f003:**
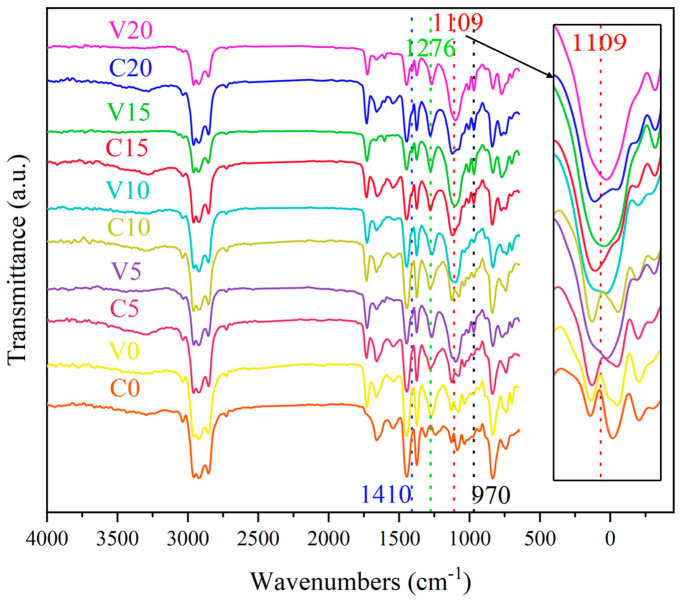
Infrared spectra of OV-POSS/NR composites.

**Figure 4 ijms-26-02001-f004:**
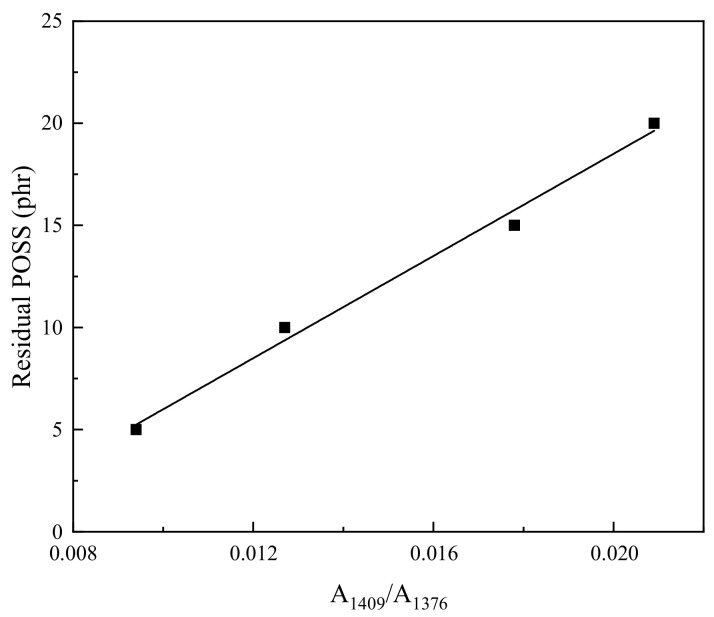
IR calibration curve for determining residual POSS contents in OV-POSS/NR composites.

**Figure 5 ijms-26-02001-f005:**
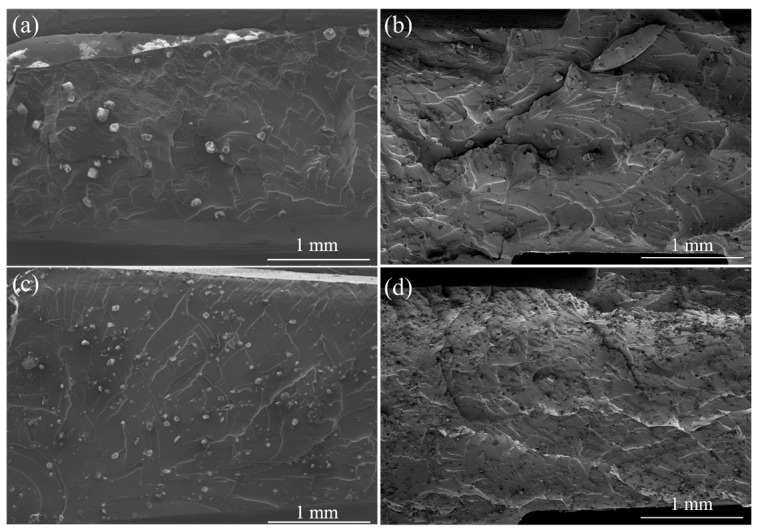
Overview scanning electron microscopy of OV-POSS/NR vulcanizates with different OV-POSS loading (**a**) 5 phr; (**b**) 10 phr; (**c**) 15 phr; (**d**) 20 phr.

**Figure 6 ijms-26-02001-f006:**
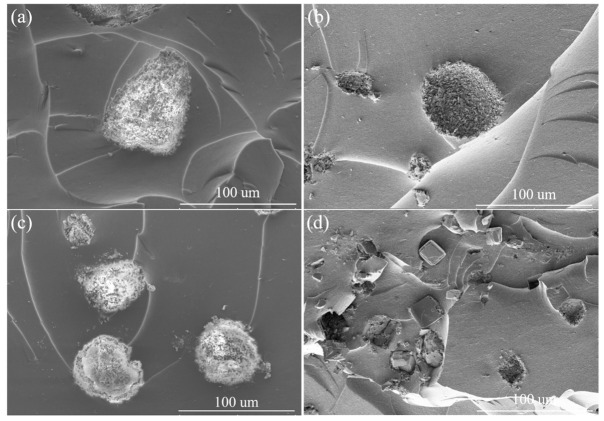
Scanning electron microscopy of the dispersed phase of OV-POSS/NR vulcanizates with different OV-POSS loading (**a**) 5 phr; (**b**) 10 phr; (**c**) 15 phr; (**d**) 20 phr.

**Figure 7 ijms-26-02001-f007:**
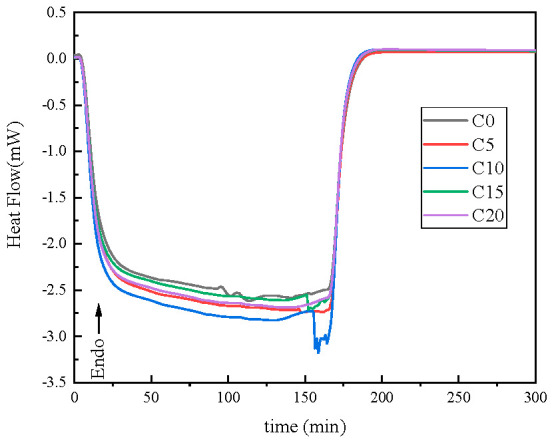
Exothermic curves of co-crosslinking reaction for OV-POSS/NR compounds.

**Figure 8 ijms-26-02001-f008:**
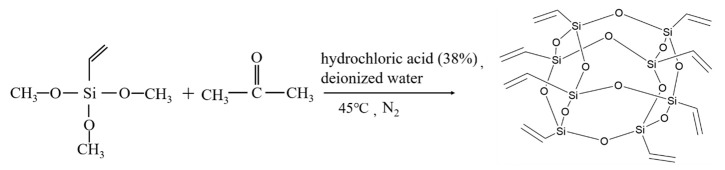
Synthesis route of OV-POSS.

**Figure 9 ijms-26-02001-f009:**
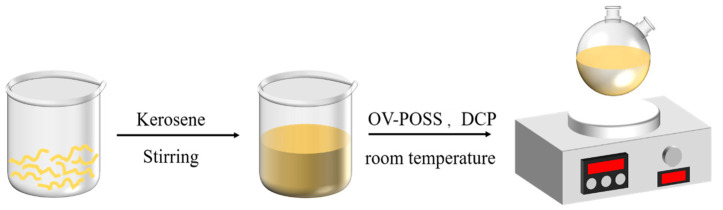
Preparation flow chart of OV-POSS/NR composites.

**Table 1 ijms-26-02001-t001:** Proportion of OV-POSS in OV-POSS/NR composites.

Sample	Feed Material Ratio (phr)	Concentration of Residual OV-POSS (mol%)	A_1410_/A_1376_	Crosslinked OV-POSS (mol%)
V0	0.00	0.00	-	-
V5	5.00	19.9	0.0060	80.1
V10	10.00	6.2	0.0057	93.8
V15	15.00	71.3	0.0137	28.7
V20	20.00	66.2	0.0157	33.8

**Table 2 ijms-26-02001-t002:** Reation enthalpy during the isothermal crosslinking reaction.

Sample	ΔH (J/g)	Content of Crosslinked OV-POSS (phr)
C0	2046	0
C5	2386	4.0043
C10	2862	9.3796
C15	2208	4.3050
C20	2368	6.7575

## Data Availability

The data that support the findings of this study are available from the corresponding author upon reasonable request.

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
