# Peer review of "Effect of Co-Crosslinking Reaction on the Morphology of Octavinyl Polyhedral Oligomeric Silsesquioxane/Natural Rubber Composites"

_ijms, 2025, doi:10.3390/ijms26052001_

Round 1
Reviewer 1 Report
Comments and Suggestions for Authors
The manuscript, "Effect of co-crosslinking reaction on the morphology of OV-POSS/NR composites", details the synthesis and characterization of octavinyl-polyhedral oligomeric silsesquioxane and further, their blending with natural rubber. In the reviewer’s opinion, the work is poorly presented with incomplete characterization and discussion.
- The title and introduction do not complement. Why is the focus first on natural rubber?
- The title mentions ‘morphology’; however, the authors report only SEM images as results for the surface features.
- There is no need of Figure 2 in face of the descriptive figure 3.
- In figure 3, the authors should provide an inset for Si-O-Si characteristics frequencies. In the main figure it is very difficult to discern the comparative changes.
- There is a prominently broad peak in all the C samples from approx. 3200 – 3450 cm-1 which the authors haven’t discussed.
- A lot of words throughout the text have been misspelt e.g. instensity and domanins
- The section 2.4 is difficult to comprehend and the description seems inadequate.
Comments on the Quality of English Language
The English needs to be carefully checked.
Reviewer 2 Report
Comments and Suggestions for Authors
This is a short experimental paper on the introduction of OV-POSS into NR. A range of concentration was covered (0-20 phr), but limited characterization was performed. The scope of the study is also very limited: what is the purpose/application of the final materials ?
Other corrections to make are:
Three keywords is not enough.
Figure 1 needs a title with units for the vertical axis.
Line 78: change the last “10” for “20”.
Figure 7 needs a title with units for the vertical axis.
Figure 7 and Table 2 must report the testing temperature.
Table 2: put only 2 decimals for all values of the last column. Any experimental error (number of repetition) ?
Line 183: superscript for “o”.
Line 185: superscript for “o”.
For FTIR, how many spectra were used to get the average ?
Line 222: superscript for “o”.
Most of the references are incomplete: missing page/paper number.
Also, the written English quality can be improved at several places (choice or words)…
Comments on the Quality of English LanguageSome parts (such as the Abstract and Introduction) must be improved.
Reviewer 3 Report
Comments and Suggestions for Authors
The study examines the effect of co-crosslinking reactions on the morphology of OV-POSS/NR composites, focusing on how OV-POSS incorporation influences dispersion and crosslinking behavior. Additionally, the research contrasts these findings with previously studied sulfur-cured systems.
To enhance the clarity and publishability of this paper, please address the following points:
- What do the numbers in C0–C20 and V0–V20 represent?
- In Table 1, why does the 15% sample exhibit the lowest crosslinked OV-POSS content?
- In Table 2, the observed trend appears inconsistent—could you provide a brief explanation?
- In Section 2.4, the DSC values are attributed solely to reaction enthalpy. However, phase separation and crystallization can also contribute to energy changes. How were these effects ruled out?
- Additionally, given your discussion on crosslinking and nanoparticle reinforcement, I suggest citing the following paper: Kanbargi et al., "Amplifying Nanoparticle Reinforcement through Low Volume Topologically Controlled Chemical Coupling," ACS Macro Letters, 2024. This work explores controlled chemical coupling to enhance nanoparticle reinforcement, which may provide additional context for your findings.
Round 2
Reviewer 1 Report
Comments and Suggestions for Authors
The revised version is sufficient
Author Response
Comments 1:The revised version is sufficient
Response 1:Received, thank you.

Reviewer 3 Report
Comments and Suggestions for Authors
The list of references is missing from this manuscript.
Author Response
It may be due to the problem of the submission system that the list of references disappeared. The revised manuscript opened on my client can see the list of references, and I am very sorry for this. I resubmit a revised manuscript.
